# Compact and Broadband Microstrip Band-Stop Filters with Single Rectangular Stubs

**Yusuke Kusama [1],\* and Ryota Isozaki [2]**

[1]   National Institute of Technology, Kagawa College, Kagawa 769-1192, Japan
[2]   Advanced course, National Institute of Technology, Kagawa College, Kagawa 769-1192, Japan;
      a17502@sr.kagawa-nct.ac.jp
\*   Correspondence: kusama@cn.kagawa-nct.ac.jp; Tel.: +81-875-83-8544

**Abstract:** In this research, a compact and broadband microstrip line quarter-wavelength open circuited stub, which is the core of the band-stop filter, is studied from the viewpoint of the characteristic impedance ratio between the main transmission line and the stub line. Furthermore, a circuit pattern in which an inductive diaphragm is inserted at the stub attachment point using a stepped impedance structure is examined, and an evaluation of frequency adjustment and miniaturization is investigated. The results are compared with the well-known radial stub. Good agreement was obtained between the measured and simulated values up to 5 GHz. Good stop bandwidth was obtained, and the validity of the proposed method is confirmed. The application to other frequency bands is straightforward. The proposed structure is applicable as an alternative to the existing radial stub used for bias *T* to prevent the reverse flow of the Radio frequency (RF) signal to direct current (DC) source. It is also applicable for the waveguide E-plane band-stop filter, for preventing unwanted leakage from narrow gaps by substituting to a short-circuited stub with a capacitive window, by using the same approach used in the microstrip line H-plane discontinuity.

**Keywords:** microwave; filter; microstrip; bandstop filter; quarter-wavelength resonator

## 1. Introduction

With the development of RF technology, RF products and services spreading in modern society have been advanced and diversified. In addition, the amount of expert knowledge and technical ability required by engineers is increasing, and the fundamental things listed in standard textbooks are not enough for practical use [1,2]. In order to foster technicians who can respond to the needs of modern society, sophisticated educational research programs to use the knowledge of pioneers to understand advanced expertise are essential to meet the demands of this era [3,4]. RF components like filters are important components supporting the above system.

In this study, we focus on a microstrip line (MSL) band-stop filter (BSF) by using a quarter-wavelength open circuited stub. The basic structure, in which the stub line width is equal to the signal line width, is simple and easy, but it is a narrow bandwidth and has poor practicality for BSF. As a general trend for MSL filters, compact and broadband characteristics are desirable, in addition to easy fabrication. To construct a band-stop filter, the following two methods are conceivable in a two-terminal network. One is to shunt a series resonance circuit to the transmission line, and the other is to insert a parallel resonance circuit in series in the transmission line. In any case, the transmission zeros are obtained at quarter-wavelength resonance. For example, corrugation or periodic structure like a photonic band gap (PBG) can obtain the highest attenuation in a wide band [5–7]. Cascading or multistage by filter synthesis is effective to achieve the desired stop bandwidth [8,9]. The butterfly structure using symmetry is effective for broadening the bandwidth, compared with the single

structure [10–13]. However, in these examples, there is a feature that the circuit pattern becomes large. To overcome this problem, some methods for compatibility between compactness and broadband by a combination of existing structures, such as spureline, interdigital, and the quarter-wavelength open circuited stub, have been reported [14–23]. However, the bandwidth becomes narrower because of the trade-off relationship between the size reduction and broadband. In addition, it is inevitable that the manufacturing process becomes troublesome because of the miniaturized structure.

As another technique to increase the compatibility between compactness and broadband, we propose to increase the stop bandwidth as much as possible, for a simple single rectangular stub with frequency adjustment by inserting a reactance element at the attachment point of the stub. The method of widening the stop bandwidth by using a low impedance line for the stub is well known, but there are few quantitative reports on the case where the line widths of the main transmission line and the stub are different. In addition, we propose a method to adjust the resonant frequency by inserting an inductive window at the stub attachment point, not changing the resonator size. As a similar structure, radial stubs are widely used, because of their compactness and broadband, but the theoretical analysis is somewhat complicated, and the principle is not simple, except that the effective attachment point of the stub is identified [24].

In this research, firstly we study a rectangular quarter-wavelength open stub from the viewpoint of the characteristic impedance ratio between the main transmission line and the stub line. Secondly, we investigate the possibility for miniaturization by series-connecting reactance to the attachment point of the stub. Furthermore, we compare the results with the well-known radial line stub, to confirm the validity we propose. Section 2 describes the principle of broadband and miniaturization. Section 3 describes the measurement sample, and Section 4 discusses the measurement results. Finally, we summarize in Section 5.

## 2. Theoretical Calculation

### 2.1. Principle of the Broadband Stub

Figure 1 shows a circuit model of the open circuited stub. In the schematic view in Figure 1a, the line width of the main transmission line is $W_0$, the line length is $l_0$, the characteristic impedance is $Z_0$, the line width of the stub line is $W_1$, the stub length is $l_1$, and the characteristic impedance is $Z_1$. Figure 1b shows the TEM transmission line model of the open circuited stub. The left side shows the equivalent circuit of the open circuited stub, attached parallel to the I/O transmission line, and the right side shows the equivalent lumped element model of the open circuited stub for $l_1 < \lambda/4$. The normalized admittance $y = Y_{\text{in}}/Y_0$ of the open circuited stub line is given by

$$y = \frac{Y_{in}}{Y_0} = jb = j\frac{Z_0}{Z_1}\tan\beta_1 l_1 \tag{1}$$

The circuit of Figure 1a is a three-stage circuit; the transmission coefficient $S_{21}$ of this circuit is given by the following equation, using the normalized admittance $y$.

$$S_{21} = \frac{2}{2+y} \tag{2}$$

Equation (2) is a complex number, but by rationalizing it to obtain its size, Equation (3) is obtained. By substituting susceptance $b$ in Equation (1) into Equation (3), then $|S_{21}|$ can be found.

$$|S_{21}| = \frac{2}{\sqrt{4+b^2}} \tag{3}$$

In order to broaden the stop bandwidth near the quarter-wavelength resonance without multistage or corrugation, it is necessary to enlarge the normalized susceptance $b$ in Equation (3). The significant parameter decides if it is the ratio of the characteristic impedance $Z_1/Z_0$ between the stub line and the

main transmission line. As an example, the center frequency 2.5 GHz in the ISM band is selected, but the proposed structure is not limited to this frequency band. Application to other frequency bands is straightforward. Figure 2 shows $|S_{21}|$ and $|S_{11}|$ calculated values when $Z_1/Z_0$ is 1/2, 1, and 2. It can be seen that the bandwidth widens if $Z_1/Z_0$ is smaller. In this study, the characteristic impedance of the main transmission line is fixed to $Z_0 = 50\ \Omega$, so that the impedance matching with the measurement system. The stub becomes a low impedance line with $W_1 > W_0$ for $Z_1/Z_0 < 1$, and becomes a high impedance line with $W_1 < W_0$ for $Z_1/Z_0 > 1$. On the other hand, the bandwidth can be explained from quality factor $Q$ at the quarter-wavelength series resonance [25].

$$Q = \frac{\omega_0 L}{R} = \frac{1}{\omega_0 C R} \tag{4}$$

Increasing the stop bandwidth is the same as the decrease of $Q$, so from Equation (4), it is necessary to increase $C$ and $R$, or to decrease $L$ of the resonator. Using a low impedance line with a large line width for the stub is equivalent to increasing the $C$ of the resonator from the electrostatic analysis; as a result, $Q$ decreases and becomes broadband.

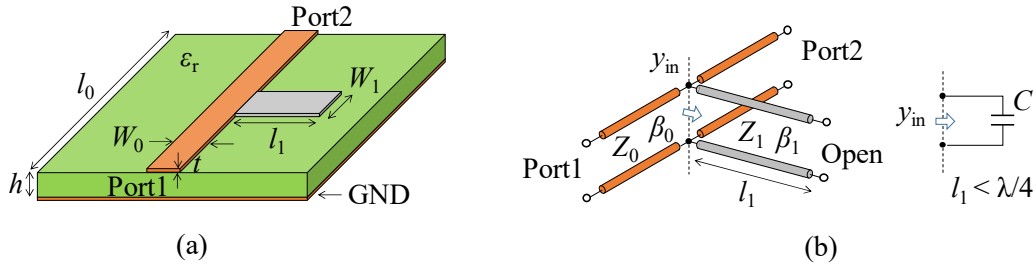

(a)                              (b)

**Figure 1.** (**a**) Schematic view of a microstrip line (MSL) open-circuited stub. (**b**) Transmission line equivalent circuit of MSL open-circuited stub, and its input admittance when $l_1 < \lambda/4$.

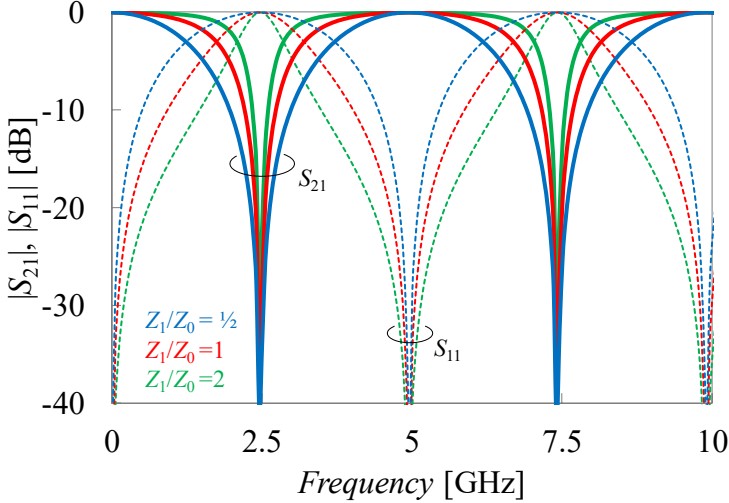

**Figure 2.** Calculated results of transmission coefficients $|S_{21}|$ and reflection coefficients $|S_{11}|$ for different characteristic impedance ratios between the stub line and the main transmission line ($Z_1/Z_0$). Three patterns show $Z_1/Z_0 = 1/2$, 1, and 2. The stop bandwidth widens if $Z_1/Z_0$ is smaller. Smaller impedance ratio means that the stub line width $W_1$ becomes wider.

## 2.2. Principle of Miniaturization by Reactance Insertion Stub

Figure 3 shows a circuit model of the open circuited stub with a stepped impedance structure. In the schematic view in Figure 3a, a reactance element, which has different line width to the stub line width, is connected to the stub attachment point in series connection. Figure 3b shows the TEM

transmission line model of the open circuited stub with a series inductance. The left side shows the equivalent circuit of the open circuited stub with series inductance attached parallel to the I/O transmission line, and the right side shows the equivalent lumped element circuit model of the open circuited stub, with a series inductance for $l_1 < \lambda/4$. Since the open circuited stub appears to be capacitive in the range of $l_1 < \lambda/4$, if the reactance is inductive, series resonance can be made with a length that is shorter than the original resonant length. Lowering the quarter-wavelength resonant frequency without increasing the resonant length leads to miniaturization. In order to represent the series inductance in the MSL, there is a method using a high impedance line with a short electric length, from the same idea as the stepped impedance structure. It is the same as making a thin constriction at the attachment point of the resonator, or as driving a rectangular wedge. In Figure 3b, the normalized input admittance $y_{in}$ when seeing open stub from the attachment point is given by Equation (5).

$$y_{in} = \frac{Y_{in}}{Y_0} = j\frac{1}{(Y_0/Y_1)\cot\beta_1 l_1 - Y_0\omega L} = jb \tag{5}$$

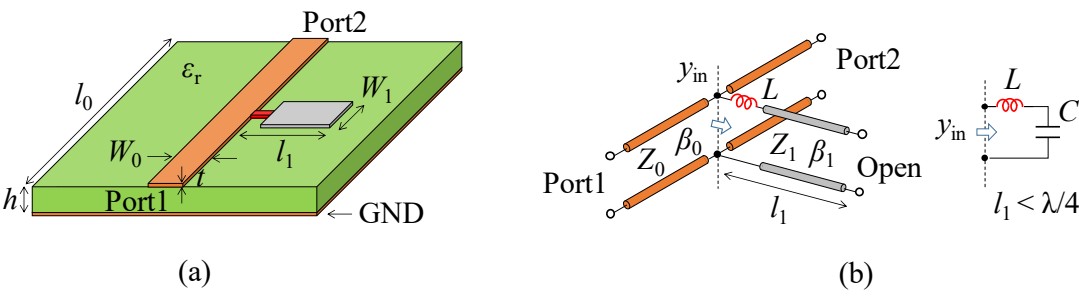

(a)                                                                                       (b)

**Figure 3.** Circuit model of an MSL open-circuited stub with inductance inserted at the attachment point to the main transmission line. (**a**) Schematic view of MSL open circuited stub with a reactive element; (**b**) equivalent circuit of MSL open-circuited stub with an inductance $L$.

Figure 4 shows the calculated results of $|S_{21}|$ and $|S_{11}|$ for the series inductance value $L = 0$ nH, 1 nH, 10 nH, and 100 nH when $Z_1/Z_0 = 1/2$. It can be seen that the transmission zero frequency decreases as $L$ increases. However, as shown in Equation (4), as $L$ increases, $Q$ increases to a narrow bandwidth. In the case of $Z_1/Z_0 < 1$, where the $C$ of the resonator is large, it becomes broadband, but the size of the resonator also becomes large. On the other hand, when the $L$ of the resonator is large, it becomes a narrow band, but the size of the resonator becomes small. Therefore, there is a trade-off relationship between miniaturization and broadband bandwidth. Radial stubs are well known as a method to achieve both compact and broadband bandwidth. In the case of radial stubs, $C$ becomes larger with the fan-shaped plate, and the characteristic impedance of the stub line decreases as the central angle becomes larger; therefore, it is considered that the principle of broadband and miniaturization is the same as the low impedance open-circuited stub with a series inductance proposed in this study.

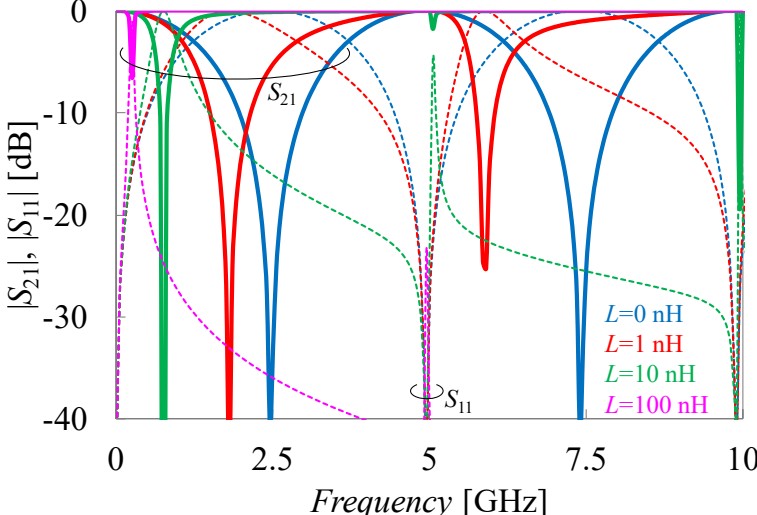

**Figure 4.** Calculated results of $|S_{21}|$ and $|S_{11}|$ for different inductances when $Z_1/Z_0 = 1/2$. Four patterns show the case $L = 0$ nH, 1 nH, 10 nH, and 100 nH.

## 3. Measurements

### 3.1. Measurement Samples

Figure 5 shows the measurement samples in this study. Patterns (a), (b), and (c) are rectangular stubs, and patterns (d) and (e) are rectangular stubs with a series reactance at the attachment point. Pattern (f) is the radial stub for comparison. The substrate used in this experiment is a glass cloth-based, epoxy resin copper clad laminate Sunhayato NZ-30KR (FR-4), which has a relative dielectric constant $\varepsilon_r = 4.4$ and dielectric loss tangent $\tan \delta = 0.02$ at 10 GHz. The substrate sizes are longitudinal width 75 mm, lateral width 100 mm, and height $h = 1.6$ mm, with a copper foil with a thickness $t = 35$ μm. All sample circuit patterns are fabricated by etching. All I/O line widths are set to $W_0 = 3$ mm, so that the characteristic impedance becomes $Z_0 = 50$ Ω for the matching to the measurement system.

In pattern (a), the characteristic impedance of the stub line is investigated on the four types of $Z_1$ = 12.5 Ω, 25 Ω, 50 Ω, and 100 Ω for $W_1 = 19.4$ mm, 8.4 mm, 3 mm, and 0.7 mm. In each, the impedance ratio is $Z_1/Z_0 = 1/4$, 1/2, 1, and 2. As a method of further reducing the ratio of $Z_1/Z_0$, there is a method of increasing the characteristic impedance of the main transmission line. Therefore, as in pattern (b), only the junction width to the stub line is set to $Z_0 = 100$ Ω, where the impedance ratio is $Z_1/Z_0 = 1/4$. In pattern (c), a tapered line impedance transformer consisting of nine steps is used to evaluate the reflection characteristics of the stub only. Pattern (d) is an inductance expressed by a 0.7 mm length line at the stub attachment point, and its line widths are set to two types, $g = 7$ mm and 0.7 mm, in order from the smallest inductance. Pattern (e) is made by a longer line length $n$ in order to further increase the inductance of the pattern (d), and it was set to two types of $n = 4$ mm and 8 mm, in order from the smallest inductance. By setting the slit in the input side like (e), $L$ is increased without changing the size of the resonator. Pattern (f) is a radial stub, and two types of central angles $\alpha = 60°$ and 90° are examined. In the case where the central angle is $\alpha = 60°$, the inner radius is $R_1 = 1.7$ mm and the outer radius is $R_2 = 13.1$ mm. And in the case of the central angle is $\alpha = 90°$, $R_1 = 2.1$ mm and $R_2 = 12.9$ mm are used for resonating at $f = 2.5$ GHz by using the empirical formula [24].

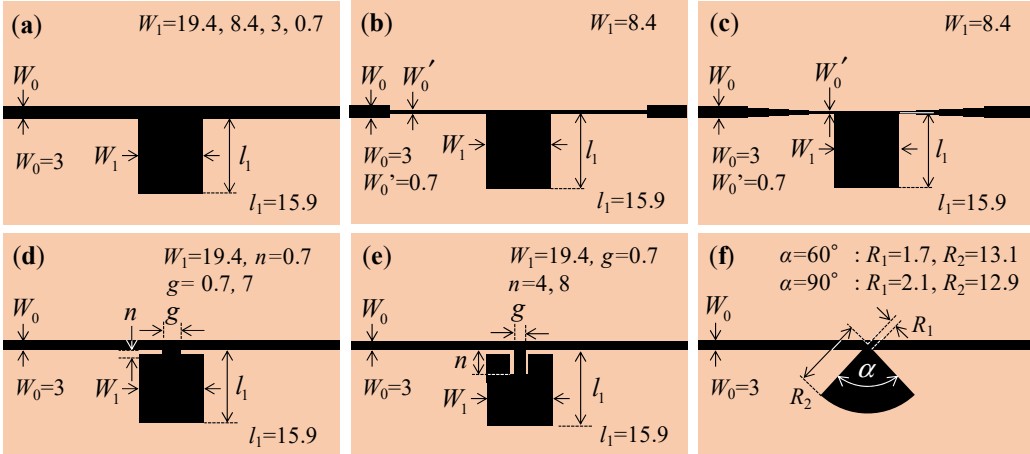

**Figure 5.** Schematic view of measurement samples examined in this study. All numerical units are in mm. (**a**) Rectangular stubs of four different characteristic impedance ratios $Z_1/Z_0$ = 1/4, 1/2, 1, and 2. (**b**) Rectangular stub of $Z_1/Z_0$ = 1/4, with no matching section except for the either end of the I/O line. (**c**) Rectangular stub of $Z_1/Z_0$ = 1/4 with a tapered line. (**d**) Rectangular stubs of $Z_1/Z_0$ = 1/4 with two different series inductances. (**e**) Rectangular stubs of $Z_1/Z_0$ = 1/4 with a larger series inductance. (**f**) Radial stubs of central angle $\alpha$ = 60° and 90°.

## 3.2. Measurement System

Figure 6 shows the measurement system. The tip of the 3.5 mm coaxial connector was set on the calibration plane, measurement samples were inserted, and the transmission characteristic $S_{21}$ and reflection characteristic $S_{11}$ were measured in the range of 0.01–10 GHz. The instrument used was a vector network analyzer Agilent E8362C, the IF bandwidth is 1 kHz, and the number of measurement points is 401. In addition, SMA connectors RS 526-5785 are attached to both ends of the substrate for connection with the coaxial calibration plane.

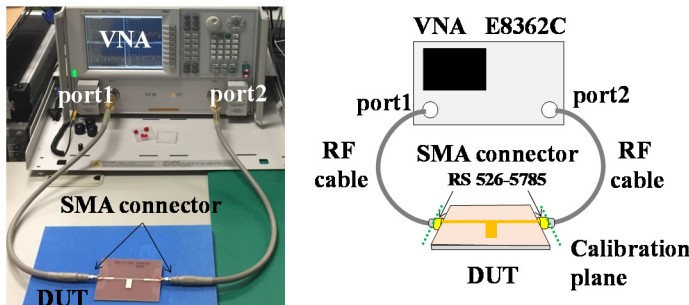

**Figure 6.** Measurement system by using a vector network analyzer (VNA). The device under test (DUT) is inserted between the short, open, load, and through (SOLT) calibration planes, not including SMA connectors attached to the DUT substrate.

## 4. Results and Discussion

### 4.1. Validation of Measurement Results

As an example, Figure 7 shows the transmission and reflection characteristics $|S_{21}|$ and $|S_{11}|$ when the impedance ratio is $Z_1/Z_0$ = 1/2, in the case of the sample pattern (a) in Figure 5. The three types of data show the calculated values based on the TEM transmission line theory, the EM simulated value by using a commercial EM simulator Ansys HFSS v.11, and measured values. There exists a discrepancy between the theoretical calculation and the other two above 4 GHz. This is because, for the sake of simplicity, the theoretical calculation does not consider the dielectric loss and the conductor loss due to the skin effect. Also, the EM simulated $|S_{21}|$ value is larger than the measured value above

6 GHz, despite the solve-inside-conductor option. It is considered that the conductor loss due to the skin effect at higher frequency is not calculated with sufficient accuracy in the HFSS [26]. Therefore, in this study, it is confirmed that measurements can be made with sufficient accuracy up to 6 GHz. Therefore, the transmission characteristics of 0–5 GHz, at which the quarter-wavelength resonance is observed, will be compared.

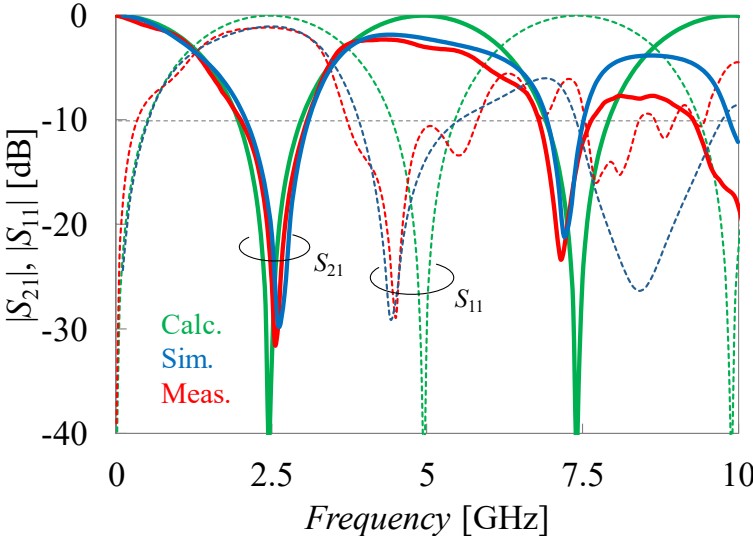

**Figure 7.** Compared results of $|S_{21}|$ and $|S_{11}|$ in the case of the rectangular stub when $Z_1/Z_0 = 1/2$. Three kinds of data show the calculated values based on the TEM transmission line theory and the EM simulated value by using an Ansys HFSS simulator and measured values. Good agreement is obtained up to 5 GHz for the measured value and the simulated value.

### 4.2. Bandwidth Change of Rectangular Stubs

Figure 8 shows the measurement results of $|S_{21}|$ and $|S_{11}|$ in the case of rectangular stubs. Four kinds of data indicated by line (a) show the case where the impedance ratio is $Z_1/Z_0 = 2$, 1, 1/2, and 1/4 in the pattern (a) shown in Figure 5. The other two kinds of data indicated by lines (b) and (c) show the cases of $Z_1/Z_0 = 1/4$ in pattern (b) and (c) shown in Figure 5. In the general EMC design, the threshold level of the transmission is often set to $T = -30$ dB, which can obtain sufficient shielding. However, since it is difficult to measure the bandwidth attenuated to $-30$ dB, we set the threshold as $T = -10$ dB. Table 1 shows the change of the bandwidth with respect to the threshold and the quarter-wavelength resonance point. The numerical values in parentheses represent the bandwidth expansion ratio and the increase/decrease ratio of the quarter-wavelength resonance point, with $Z_1/Z_0 = 1$ as a reference. In the case of pattern (a), it shows that by setting $Z_1/Z_0$ from 1 to 1/2, the stop bandwidth at each threshold increases by 1.8 times. From this result, it is also found that by increasing $Z_1/Z_0$ from 1 to 1/4, the stop bandwidth increases 2.9 times. On the other hand, for patterns (b) and (c), the bandwidth increases 3.6 times. It was found that the taper deteriorates the stop band characteristics on the lower frequency side. It was also confirmed that the bandwidth of patterns (b) and (c) is wider than that of the rectangular stub of the pattern (a), even with the same characteristic impedance ratio. Furthermore, as the value of $Z_1/Z_0$ decreases, the capacitance $C$ of the circuit increases, and it is confirmed that $Q$ decreases, as shown by the Equation (4). It should be noted that the reason why the quarter-wavelength resonance point shifts is considered to be that the propagation constant $\beta_1$ changes as the stub line width $W_1$ increases. From the above, it is found that the transmission zero frequency shifts to higher frequency, although the band is wider as $Z_1/Z_0$ becomes smaller.

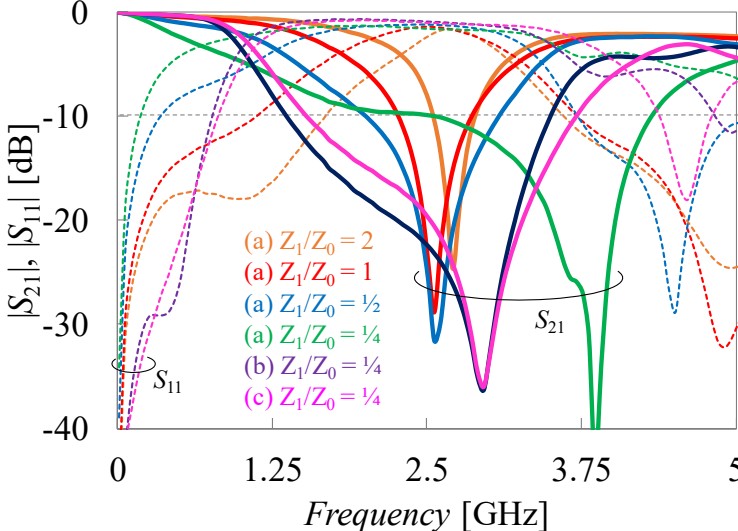

**Figure 8.** Measured $|S_{21}|$ and $|S_{11}|$ results of rectangular stubs, with different impedance ratios $Z_1/Z_0$ as shown in Figure 5a–c.

**Table 1.** Stop bandwidths in the case of rectangular stubs.

| Circuit Pattern | $Z_1/Z_0$ | $-10$ dB BW [GHz] ([1]) | $f_0$ [GHz] ([1]) |
|:---:|:---:|:---:|:---:|
| (a) | 2 | 0.35 (0.6) | 2.71 (1.1) |
| (a) | 1 | 0.60 (1.0) | 2.56 (1.0) |
| (a) | 1/2 | 1.05 (1.8) | 2.56 (1.0) |
| (a) | 1/4 | 1.75 (2.9) | 3.86 (1.5) |
| (b) | 1/4 | 2.15 (3.6) | 2.96 (1.2) |
| (c) | 1/4 | 2.15 (3.6) | 2.96 (1.2) |

[1] Indicate the magnification for $Z_1/Z_0 = 1$.

### 4.3. Resonance Frequency Change of the Reactance Insertion Stub

Figure 9 shows the measurement results of $|S_{21}|$ and $|S_{11}|$ of the rectangular stub with a series inductance. The three types of data indicated by line (d) show the case where the thickness of the diaphragm is fixed at $n = 0.7$ mm, and the aperture width is changed to $g = 19.4$ mm, 7 mm, and 0.7 mm in pattern (d) in Figure 5. It should be noted that the case $g = 19.4$ mm means that there is no diaphragm, which is equivalent to the case of $Z_1/Z_0 = 1/4$ of pattern (a) in Figure 5. The two types of data indicated by line (e) show the cases where the aperture width is fixed at $g = 0.7$ mm, and the diaphragm thickness was changed to $n = 4$ mm and 8 mm at pattern (e) in Figure 5. It can be seen that the transmission zero frequency is shifted to the left as the aperture width $g$ is narrowed. Since narrowing the aperture width is equivalent to increasing $L$ of the circuit, it can be seen that the transmission zero frequency can be lowered without changing the resonance length, so that the size can be reduced. Furthermore, it can be seen that as the diaphragm thickness $n$ is increased, the transmission zero frequency is shifted further to the left. As $n$ increases, it is equivalent to increasing the value of $L$. It is also found that as the inductance becomes larger, the size of the resonator can be reduced. Table 2 shows the change of the bandwidth with respect to the threshold $T = -10$ dB and the change of the transmission zero frequency $f_0$.

The numerical values in parentheses represent the bandwidth expansion ratio and the increase/decrease ratio of the transmission zero frequency, compared with the case $Z_1/Z_0 = 1$ as a reference. From the table, it can be seen that the narrower the aperture width, the narrower the bandwidth. As can also be seen from $Q$ in Equation (4), the larger the $L$, the larger the $Q$ value, and the sharper the resonance is. From the above results, it is found that there is a trade-off relationship between miniaturization and broadband.

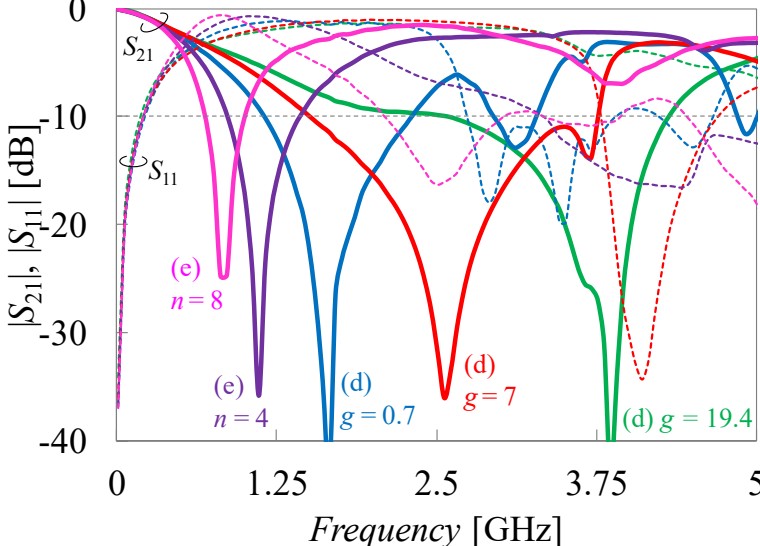

**Figure 9.** Measured $|S_{21}|$ and $|S_{11}|$ results of rectangular stubs of $Z_1/Z_0 = 1/4$ with different series inductances, as shown in Figure 5d.

**Table 2.** Stop bandwidth in the case of rectangular stubs with the series inductance.

| Circuit Pattern | $Z_1/Z_0$ | $-10$ dB BW [GHz] ($^1$) | $f_0$ [GHz] ($^1$) |
|---|---|---|---|
| (d) $g = 19.4$, (a) | 1/4 | 1.75 (2.9) | 3.86 (1.5) |
| (d) $g = 7.0$ | 1/4 | 2.25 (3.7) | 2.56 (1.0) |
| (d) $g = 0.7$ | 1/4 | 1.10 (1.8) | 1.66 (0.6) |
| (e) $n = 4.0$ | 1/4 | 0.55 (0.9) | 1.11 (0.4) |
| (e) $n = 8.0$ | 1/4 | 0.30 (0.5) | 0.86 (0.3) |

$^1$ Indicate the magnification for $Z_1/Z_0 = 1$.

### 4.4. Comparison with Radial Line Stub

Figure 10 shows $|S_{21}|$ and $|S_{11}|$ measurement results of a well-known radial stub. Two types of data indicated by line (f) show the case of the center angle $\alpha = 60°$ and $90°$ in the pattern (f) in Figure 5. For comparison, the results of the reactance insertion stub in the case where $Z_1/Z_0 = 1/4$ with $g = 7$ mm, which has the same center frequency at 2.5 GHz, are shown by line (d). It is confirmed that the larger the central angle $\alpha$, the wider the bandwidth is. Table 3 shows the bandwidth change and the transmission zero frequency with respect to the threshold level $T = -10$ dB. The numerical values in parentheses represent the bandwidth expansion ratio and the increase/decrease ratio of the transmission zero frequency, compared with the case $Z_1/Z_0 = 1$ as a reference. It is found that the stop bandwidth increases by 2.0 times when $\alpha = 60°$, and increases by 2.4 times when $\alpha = 90°$. Although it is already known that the band-stop characteristics improve when $\alpha$ is increased, it is considered that $C$ of the resonator increases. The proposed structure has a similar transmission characteristic to the radial stub, but the bandwidth increases to 3.7 times wider than the reference value. It is also confirmed that the operating principle of the radial stub is equivalent to the circuit model, with reactance inserted in the low impedance stub described in Section 2. The characteristic impedance ratio of the reactance insertion stub used in this study is the case $Z_1/Z_0 = 1/4$, but by using a stub with a smaller impedance ratio, there is a possibility to obtain much broader band-stop characteristics than the radial stub.

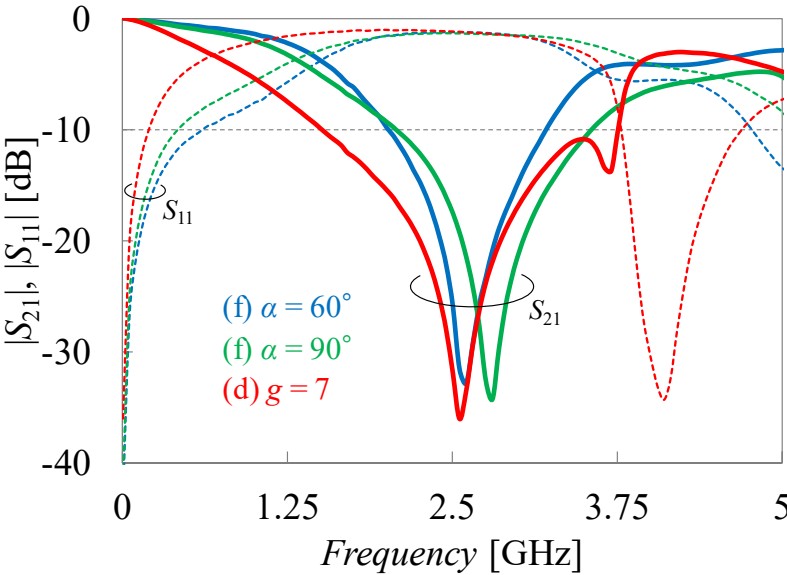

**Figure 10.** Measured $|S_{21}|$ and $|S_{11}|$ results of rectangular stubs of $Z_1/Z_0 = 1/4$ with a series inductance $g = 7$ mm, as shown in Figure 5d, compared with two radial stubs.

**Table 3.** Stop bandwidth in the case of radial stubs.

| Circuit Pattern | $\alpha$ [Degree] | $-10$ dB BW [GHz] ($^1$) | $f_0$ [GHz] ($^1$) |
|:---:|:---:|:---:|:---:|
| (f) | $\alpha = 60°$ | 1.20 (2.0) | 2.61 (1.0) |
| (f) | $\alpha = 90°$ | 1.45 (2.4) | 2.81 (1.1) |
| (d) $g = 7$ | $Z_1/Z_0 = 1/4$ | 2.25 (3.7) | 2.56 (1.0) |

$^1$ Indicate the magnification for $Z_1/Z_0 = 1$.

## 5. Conclusions

In this study, we examined compact and broadband microstrip band-stop filters with single rectangular stubs. First, in order to confirm the principle of broadening the bandwidth, we focused on the characteristic impedance ratio between the stub line and the main transmission line. The change in bandwidth in rectangular stubs was examined, and we confirmed that the bandwidth widened as $Z_1/Z_0$ became smaller. However, it was found that the transmission zero frequency shifts, due to the change in the propagation constant of the stub line, and this tendency appeared as a remarkable change in the case where $Z_1/Z_0$ is smaller. Also, it was found that even if the impedance ratio is the same, if the width of the main transmission line was narrowed, the change of the transmission zero frequency could be suppressed to be smaller.

Next, in order to confirm the principle of frequency adjustment and miniaturization, the case of inductance element $L$ using a stepped impedance structure, which plays a role of lowering the transmission zero frequency, was examined, and was connected in a series to the attachment point of the stub. From this study, it was found that the transmission zero frequency decreases as $L$ increases. However, as $L$ became larger, the $Q$ became larger and the bandwidth narrowed, so it has turned out that there is a trade-off relationship between miniaturization and broadband bandwidth.

In addition, when comparing these results with the band-stop characteristics of the well-known radial stub, they were found to agree well with the circuit pattern with appropriate $L$ inserted in the low impedance stub. Then it was confirmed that the operating principle of the radial line stub is equivalent to the circuit pattern in which the reactance was inserted into the low impedance stub. Therefore, the validity of increasing the stop bandwidth as much as possible for a single rectangular stub with frequency adjustment, by inserting a reactance element at the attachment point of the stub, was confirmed. Although not described in this report, it is possible to obtain a broader band

characteristic than the radial line stub by using a stub with a smaller impedance ratio. In the future, based on the results of this study, we plan to investigate the broadband and miniaturization of the E-plane short-circuited stub, like the choke structure inserted in the waveguide.

**Author Contributions:** Conceptualization, methodology, writing (review and editing), investigation, validation, visualization, supervision, and project administration were contributed to by Y.K.; data curation, formal analysis, writing (original draft preparation), and validation contributed by R.I.

**Funding:** This research received no external funding.

**Conflicts of Interest:** The authors declare no conflict of interest.

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
