# Peer review of "Compact and Broadband Microstrip Band-Stop Filters with Single Rectangular Stubs"

_applsci, doi:10.3390/app9020248_

Round 1

Reviewer 1 Report

1-The paper is well written, but the novelty isn't clear. However, I couldn't find any novelty in this paper. The concepts presented in this paper are well known for decades in the microstrip filter designing.

2-The calculation in this way is something well-known, all schematic previously presented before.

It seems section 2.2 is the core idea of the paper, however, adding design parameter as the a isn't something novel.\

3-However, I suggest this paper for major revision, the authors might clear distinguished what is the novelty of their research. I think the idea for miniaturizing the filter size based on impedance calculation is interesting. But, the authors should use it to design more complicated filter geometry and proof their calculation can bring miniaturization in comparison to other presented filter geometry. 

This paper might be useful for the authors:

[1] MN Jahromi , "Wide stopband compact microstrip lowpass filter using circular ring resonator and split ring resonators", Microwave and Optical Technology Letters, 2011

In this paper, the compact wide stopband filter is presented by designing the equivalent circuit and resonance frequency of circular stub calculation.

I would like to see if the authors can apply their idea on such geometry and it can be publishable and high impact paper for the readers.

Author Response

1-The paper is well written, but the novelty isn't clear. However, I couldn't find any novelty in this paper. The concepts presented in this paper are well known for decades in the microstrip filter designing.

Response 1: Since this study aims at broadband bandstop characteristics with only a single stub, it differs greatly from other documents in that no prototype based on filter synthesis theory is used. That is, the purpose of this study is to expand the bandwidth of a single stub to the limit. The suggested reference the reviewer taught us uses 3-stage LPF prototype and is not comparable with this research because it seems to be necessary to compare under the condition that the filter order is the same. Even with the stub proposed in this research, if it is redesigned by using filter synthesis theory, it should be able to obtain much broadband characteristics, but it is a secondary result and not the subject of this study. The past research which seems to be the closest to this study within the range examined by the authors is a single radial stub. However, mode analysis using Bessel function is essential for this theoretical analysis, and although the design formula based on the empirical formula is given, its operating principle is not easy. On the other hand, our study can explain the principle of operation only with the simple TEM transmission line theory, and it realizes 1.5 times the broadband characteristics than the conventional radial stub from the measurement results. Although it is possible to control the resonant frequency by inserting the inductance, at present, it is only adjustment by using electromagnetic field simulator, and the viewpoint of extraction of element value and suggestion of design formulation is a future task. A comment related to this has been added to the lines 56-57 and 52.

2-The calculation in this way is something well-known, all schematic previously presented before.

It seems section 2.2 is the core idea of the paper, however, adding design parameter as the a isn't something novel.

Response 2: In this context, the operation principle of this filter is conceptually explained by simple TEM transmission line theory, and there is no novelty in the calculation method itself. The parameter "a" is introduced as a value proportional to the inductance L for convenience. What we want to tell here is that the resonant frequency can be lowered without enlarging the resonator size and it has the same meaning as miniaturization. However, as pointed out, the introduction of the parameter "a" may be misunderstood as insisting on novelty, so we used the value of inductance L as a parameter instead. The changed parameters are L = 0, 1, 10 and 100 nH. Comments related to this have been added to lines 128-129 and 131-132.

3-However, I suggest this paper for major revision, the authors might clear distinguished what is the novelty of their research. I think the idea for miniaturizing the filter size based on impedance calculation is interesting. But, the authors should use it to design more complicated filter geometry and proof their calculation can bring miniaturization in comparison to other presented filter geometry. 

This paper might be useful for the authors:

[1] MN Jahromi , "Wide stopband compact microstrip lowpass filter using circular ring resonator and split ring resonators", Microwave and Optical Technology Letters, 2011

In this paper, the compact wide stopband filter is presented by designing the equivalent circuit and resonance frequency of circular stub calculation.

I would like to see if the authors can apply their idea on such geometry and it can be publishable and high impact paper for the readers.

Response 3: We summarize the answers in the above Answer 1 because the answer to this question seems to overlap with Question 1.

Reviewer 2 Report

The authors present a study of a microstrip line bandstop filter based on a quarter wavelength open circuit stub. Based on an initial investigation, they propose to modify the insertion point of the stub to increase the bandwidth of operation while maintaining compact device footprint and fabrication error tolerant formfactor. In particular, they insert a series reactance for this purpose. Next, they fabricate and measure their proposed filters. Finally, they compare their results to the popular radial stub obtaining satisfactory performance. 

This is a well thought out manuscript and I recommend that it be accepted after minor revisions.

Major comments:

.

1. The authors state on line 192, "This is considered that the conductor loss due to the skin effect at higher frequency is not calculated with sufficient accuracy in HFSS". 

Can the authors run any auxiliary simulations to confirm this?

2. Can the authors comment on how suitable their proposed modified stub is for designing MSL BSFs that target a particular resonance frequency?

3. Can the authors comment on how their new stub design affects the filter roll-off and how this may affect specific applications?

Minor comments:

1. Please change "MSL" in the abstract to "microstrip line". 

Author Response

Response to Reviewer 2 Comments

The authors present a study of a microstrip line bandstop filter based on a quarter wavelength open circuit stub. Based on an initial investigation, they propose to modify the insertion point of the stub to increase the bandwidth of operation while maintaining compact device footprint and fabrication error tolerant formfactor. In particular, they insert a series reactance for this purpose. Next, they fabricate and measure their proposed filters. Finally, they compare their results to the popular radial stub obtaining satisfactory performance. 

This is a well thought out manuscript and I recommend that it be accepted after minor revisions.

Major comments:

.1. The authors state on line 192, "This is considered that the conductor loss due to the skin effect at higher frequency is not calculated with sufficient accuracy in HFSS". 

Can the authors run any auxiliary simulations to confirm this?

Response 1: We added a new reference [26] since we have already studied about this point. In that study, it was known that the conductor loss due to the skin effect exceeds the dielectric loss above 6 GHz. When the conductor loss becomes so large that it can not be ignored, the characteristic impedance becomes a complex number. As a result, not only loss increases but also the mismatch occurs. The measurement results support this consideration. We know that the effect of the skin effect is obvious, but this mechanism also includes future research topics. As pointed out, the HFSS analysis conditions were reviewed once again, and the convergence condition of the S parameter was tightened in addition to the solve inside conductor option, the deviation of the 3λ/4 resonance frequency became inconspicuous. However, there is a difference of 5 dB or more from the experimental value above 8 GHz for loss. Comments related to this have been added to the line 197, 199 and 368-369.

[26] Y. Kusama, Y. Yokoi, R. Johnston, ``A Study on Conductor Loss Measurement of Microstrip Line,'' IEET - International Electrical Engineering Transactions, Vol. 4 No. 1(6), pp.40-46, January - June, 2018.  (Available at the following website. http://journal.eeaat.or.th/home/index.php)

2. Can the authors comment on how suitable their proposed modified stub is for designing MSL BSFs that target a particular resonance frequency?

Response 2: In this study, as an example, the center frequency is designed to be 2.5 GHz, but there is no particular reason to stick to this frequency in the case of the H plane MSL discontinuity circuit. Application to arbitrary frequency band is straightforward. On the other hand, when applying the proposed structure to the E-plane waveguide discontinuity circuit, it has a significant meaning to the countermeasures against leaky waves from the door gap of the microwave oven at ISM 2.5 GHz band. A comment related to this has been added to the lines 22-23 and 87-89. (This duplicates the answer to the other reviewer.)

3. Can the authors comment on how their new stub design affects the filter roll-off and how this may affect specific applications?

Response 3: A concrete application example is the bias T inserted to prevent reverse flow of the RF signal. There is a merit that the stop bandwidth is wider than the existing radial stub. Furthermore, frequency adjustment can be performed by changing the width of the inductive diaphragm without changing the size of the resonator. A more practical application is considered to be the E-plane discontinuity circuit. For example, in the EMC, leakage electromagnetic waves from the gap becomes a problem. As suggested in this research, a simple RF choke structure with easy to manufacture plays a significant role as a suppression of broadband leaky waves. In addition, if we use the SIW structure by LTCC, it seems that further applications will be expanded. A comment related to this has been added to the lines 10-11 and 13-14. (This duplicates the answer to the other reviewer.)

Minor comments:

1. Please change "MSL" in the abstract to "microstrip line". 

Response 1: We fixed as you pointed out. A modification related to this has been added to the line 15.

Reviewer 3 Report

The problematic addressed in this study is still relevant. This is a good technical paper with standard designs from theoretical study, simulated and measured planar filter ( a microstrip line (MSL) bandstop filter (BSF) by using a quarter-wavelength open circuited stub).

There is adjustments that, in my opinion, would further increase the clarity and the scientific quality of this article: 

1- it would be important to argue the choice of technology choice, ie microstrip technology! My question, why do authors choose this technology rather than coplanar technology?

2- in all the figures shown, we do not see the representation of the parameters S11 or the phase of the designed filters. Is there a reason for this choice? in my opinion, it would be important to show these parameters too because we can also guess the performance of the filter designed from the reflection parameter or the phase transition to the cutoff frequency.

3- why did the authors choose to study a cut-off filter configuration rather pass band configuration? also, why did the authors target the frequency band between 1 to 5GHz? Is there an interest in working at this frequency range or a targeted application?

4- Finally, it would also be important in the perspective part, to show a potential application for the use of this type of filter.

Author Response

Response to Reviewer 3 Comments

The problematic addressed in this study is still relevant. This is a good technical paper with standard designs from theoretical study, simulated and measured planar filter ( a icrostrip line (MSL) bandstop filter (BSF) by using a quarter-wavelength open circuited stub).

There is adjustments that, in my opinion, would further increase the clarity and the scientific quality of this article: 

1- it would be important to argue the choice of technology choice, ie microstrip technology! My question, why do authors choose this technology rather than coplanar technology?

Response 1: In this study, we aim to obtain broadband bandstop characteristics with only a single stub, so we do not expect to reduce the size by surface mounting of lumped elements with good compatibility with coplanar lines. In addition, this study is an H-plane stub, and it is important to be able to apply the same design theory to the E-plane stub. In that case, the open circuited stub is replaced by the short circuited stub and the inductive window is replaced by the capacitive window. Although it is ideal to experiment with a parallel plate waveguide which is easy to analyze theoretically, it is well known to substitute H-plane discontinuity circuit problem by MSL. A comment related to this has been added to the lines 13-14.

2- in all the figures shown, we do not see the representation of the parameters S11 or the phase of the designed filters. Is there a reason for this choice? in my opinion, it would be important to show these parameters too because we can also guess the performance of the filter designed from the reflection parameter or the phase transition to the cutoff frequency.

Response 2: As you pointed out, S11 measurements have been added to all results. Comments related to this have been added to the lines 89, 104, 128, 131, 190, 203, 208, 229, 234, 256, 261 and 278.

3- why did the authors choose to study a cut-off filter configuration rather pass band configuration? also, why did the authors target the frequency band between 1 to 5GHz? Is there an interest in working at this frequency range or a targeted application?

Response 3: Using a single stub at λ/4 resonance is suitable for BSFs. However, if we focus on the passband that appears at the λ/2 resonance, it can be used as a BPFs, but it will be larger accordingly. In addition, instead of open circuited stubs, a method of inserting capacitance at the input point using a short-circuited stub is also conceivable, but it seems that merit is low because the resonator size becomes large. In this study, as an example, the center frequency is designed to be 2.5 GHz, but there is no particular reason to stick to this frequency in the case of the H plane MSL discontinuity circuit. Application to arbitrary frequency band is straightforward. On the other hand, when applying the proposed structure to the E-plane waveguide discontinuity circuit, it has a significant meaning to the countermeasures against leaky waves from the door gap of the microwave oven using the ISM 2.5 GHz band. A comment related to this has been added to the lines 22-23 and 87-89. (This duplicates the answer to the other reviewer.)

4- Finally, it would also be important in the perspective part, to show a potential application for the use of this type of filter.

Response 4: A concrete application example is the bias T inserted to prevent reverse flow of the RF signal. There is a merit that the stop bandwidth is wider than the existing radial stub. Furthermore, frequency adjustment can be performed by changing the diaphragm width without changing the size of the resonator. A more practical application is the E-plane discontinuity circuit. In the EMC, leakage electromagnetic waves from the gap becomes a problem. Especially when it is practically difficult to fill the gap like a door of the microwave oven, a simple RF choke structure as suggested in this research plays a significant role as a suppression of leaky waves. In addition, if we use the SIW structure by LTCC, it seems that further applications will be expanded. Comment related to this have been added to the lines 10-11 and 13-14. (This duplicates the answer to the other reviewer.)

Round 2

Reviewer 1 Report

The author responses and paper update satisfy my concerns.